# Oligomer Formation and Insecticidal Activity of *Bacillus thuringiensis* Vip3Aa Toxin

**DOI:** 10.3390/toxins12040274

**Published:** 2020-04-23

**Authors:** Ensi Shao, Aishan Zhang, Yaqi Yan, Yaomin Wang, Xinyi Jia, Li Sha, Xiong Guan, Ping Wang, Zhipeng Huang

**Affiliations:** 1China National Engineering Research Center of JUNCAO Technology, College of Life Science, Fujian Agriculture and Forestry University, Fuzhou 350002, Fujian, China; es776@fafu.edu.cn (E.S.); aishanz712@126.com (A.Z.); yyq961202@163.com (Y.Y.); wangyaomin8409@163.com (Y.W.); jxy17735109183@163.com (X.J.); shal@fafu.edu.cn (L.S.); 2Key Laboratory of Biopesticide and Chemical Biology (Ministry of Education), College of Plant Protection, Fujian Agriculture and Forestry University, Fuzhou 350002, Fujian, China; guanxfafu@126.com; 3Department of Entomology, Cornell University, Geneva, NY 14456, USA

**Keywords:** *Bacillus thuringiensis*, Vip3A, *Spodoptera litura*, site-directed mutagenesis

## Abstract

*Bacillus thuringiensis* (Bt) Vip3A proteins are important insecticidal proteins used for control of lepidopteran insects. However, the mode of action of Vip3A toxin is still unclear. In this study, the amino acid residue S164 in Vip3Aa was identified to be critical for the toxicity in *Spodoptera litura*. Results from substitution mutations of the S164 indicate that the insecticidal activity of Vip3Aa correlated with the formation of a >240 kDa complex of the toxin upon proteolytic activation. The >240 kDa complex was found to be composed of the 19 kDa and the 65 kDa fragments of Vip3Aa. Substitution of the S164 in Vip3Aa protein with Ala or Pro resulted in loss of the >240 kDa complex and loss of toxicity in *Spodoptera litura*. In contrast, substitution of S164 with Thr did not affect the >240 kDa complex formation, and the toxicity of the mutant was only reduced by 35%. Therefore, the results from this study indicated that formation of the >240 kDa complex correlates with the toxicity of Vip3Aa in insects and the residue S164 is important for the formation of the complex.

## 1. Introduction

The vegetative insecticidal proteins (VIPs) from *Bacillus thuringiensis* (Bt) have been used as important insecticidal proteins for control of insect pests [1,2,3]. Vip toxins are divided into four families, including Vip1, Vip2, Vip3 and Vip4 [3]. Vip1 and Vip2 proteins act as binary toxins against some species of coleopteran and hemipteran insect [4,5]. Only 1 Vip4 protein has so far been identified but shows no activity in insects [6]. Vip3 proteins contain approximately 787 amino acid residues, showing no sequence homology with Vip1, Vip2 and Vip4 proteins [3]. Vip3 proteins have a high insecticidal activity against a wide variety of lepidopteran pests [7]. Vip3A proteins do not share the binding sites with the Bt Cry proteins [8,9,10,11], so pyramiding Vip3A proteins and Cry proteins has been widely adopted in Bt-crops [12].

Although Vip3A toxins have already been applied in transgenic crops for the control of lepidopteran pests, current understanding of the mode of action of Vip3A proteins remains limited. It is commonly assumed that Vip3A toxins exert their insecticidal activity by going through a similar sequence of events as Bt Cry1A toxins [13]. So far, the structure of Vip3A toxin has not been solved. Its structural information has been derived only by *in-silico* modeling [14,15], though the structure of the Vip3B was recently reported [16]. Studies on proteolytic activation of Vip3A proteins have shown that by a proteolytical process Vip3A protoxins are cleaved to become several major fragments, generally including fragments of 62–66 kDa, 45 kDa, 33 kDa and 19–22 kDa [17,18,19,20]. The 62–66 kDa fragment from the C-terminus of Vip3A toxins has been determined to be the main product from proteolytic processing. The 45 kDa and 33 kDa fragments are products from further processing of the 62–66 kDa fragment [18]. The 19–22 kDa fragment contains the first 199 amino acids at the N-terminus of Vip3A [21]. It has been suggested that the 62–66 kDa fragment at the C-terminus in Vip3A toxin is the activated core of the toxin [22,23,24]. However, recent studies have indicated that both of the 19–22 kDa and the 62–66 kDa fragments are required for the stability and specificity of Vip3A toxins [20]. More recently, a ~340 kDa homo-tetramer, constituted by the 19–22 kDa and the 62–66 kDa fragments, has been identified from Vip3A after treatment with trypsin or insect midgut proteases [18]. However, whether the formation of this ~340 kDa homo-tetramer is essential for the insecticidal activity of Vip3A in insects remains unknown.

A recent study of Vip3Af by Ala scanning to cover 558 out of the 788 residues showed that the most Ala substitutions in Vip3Af significantly decreased the insecticidal activity, and the proteolytically processed fragments of the Vip3Af substitution mutants displayed six different patterns by SDS-PAGE analysis [14]. Further analysis indicated that Vip3Af mutants with different proteolytic patterns could form a variety of oligomeric products [21]. The substitution of the residue T167 or G168 by Ala in the predicted 19 kDa N-terminal fragment of Vip3Af did not change the proteolytic proccessing, but both substitutions significantly decreased the insecticidal activity [14]. Sequence alignments indicated that the amino acid residues from K152 to P171 are highly conserved among the Vip3A toxins [3].

*Spodoptera litura* is a polyphagous species and a major pest of many crops worldwide due to its vigorous defoliation [25]. *S. litura* is not susceptible to Bt Cry1A toxins but highly susceptible to Vip3A toxins [26,27]. In this study, we constructed Vip3Aa mutants by site directed mutagenesis and investigated the insecticidal activity of the mutants in *S. litura*. The amino acid residue S164 in Vip3Aa protein was identified to be critical for the toxicity of Vip3Aa. Investigation of the toxicity of Vip3Aa in *S. litura* by substitutions of S164 with different amino acid residues indicated that a protein oligomer formed with the 19 kDa and the 65 kDa fragments of Vip3Aa is the toxin core necessary for the insecticidal toxicity.

## 2. Results

### 2.1. Insecticidal Activity of Residue Substituted Vip3Aa Mutants Against Neonates of S. litura

The wild-type Vip3Aa protein and its mutants at K152, D154 and S164, respectively, were prepared through a glutathione S-transferase (GST) tagged protein purification system. Vip3Aa mutants with substitution of K152 or D154 with Ala were expressed as GST-Vip3Aa-K152A and GST-Vip3Aa-D154A fusion proteins. Vip3Aa mutants from substitution of S164 with Ala, Pro and Thr, respectively, were expressed as GST-Vip3Aa-S164A, GST-Vip3Aa-S164P and GST-Vip3Aa-S164T fusion proteins. The purified GST-Vip3Aa fusion proteins and the wild-type Vip3Aa (Vip3Aa-WT) were fed to neonates of *S. litura* to determine their insecticidal activity respectively. The bioassay results showed that the substitution mutations of K152A and D154A did not significantly change the toxicity of the toxin, in comparison with the wild-type Vip3Aa protein (Table 1). However, substitution of S164 with Ala or Pro completely abolished the toxicity of Vip3Aa. In contrast, substitution of S164 with the similar amino acid residue Thr only slightly reduced the insecticidal activity (35% reduction). Mortality of the control groups fed with 100 μg/mL or 250 μg/mL of purified GST tag protein were below 5% after 96 h feeding (results not shown).

### 2.2. Analysis Vip3Aa Fragments After Proteolytic Processing

To examine the difference in the proteolytic processing among the Vip3Aa-WT and three S164 mutants, each Vip3Aa protein was processed by trypsin or midgut proteases of *S. litura* and analyzed by SDS-PAGE after heat denaturation. The tryptic fragments from the three S164 mutants and Vip3Aa-WT contained major bands at 65 kDa, 35 kDa and 19 kDa and multiple weak bands from 29 kDa to 66.4 kDa (Figure 1). Vip3Aa proteins processed by midgut proteases of *S. litura* present a different pattern to the tryptic proteins. Besides the major fragments at 65 kDa, a band at 38 kDa and another at 30 kDa were observed in the midgut proteases processed Vip3Aa-WT and three S164 mutants. The band of 19 kDa was weak or invisible after in vitro proteolytic processing of Vip3Aa by the midgut proteases of *S. litura* (Figure 1). It could be observed that after proteolytic treatment with trypsin or midgut proteases, Vip3Aa-WT and three S164 mutants showed the same protein patterns. All trypsin digested Vip3Aa proteins contained a strong band at 19 kDa. Several other protein fragments were observed with the molecular weight from 14.3 kDa to 19 kDa from Vip3Aa-WT and three S164 mutants, although some bands were weak (Figure 1).

### 2.3. Analysis of Vip3Aa Protein Complexes by Native PAGE After Proteolytic Processing

The same or similar protein digestion patterns were observed by SDS-PAGE from Vip3Aa-WT and three S164 mutants after proteolytic processing by either trypsin or midgut proteases of *S. litura*. Protein fragments from trypsin- or gut proteases-processed Vip3Aa proteins were then analyzed by native PAGE to identify the protein complexes. Two similar major bands, representing the protein complex 1 and 2, were observed from Vip3Aa-WT, Vip3Aa-S164T, Vip3Aa-S164A and Vip3Aa-S164P. However, a band, representing the protein complex 3, with a higher molecular weight than the two bands were observed from the trypsin-processed Vip3Aa-WT, Vip3Aa-S164T and Vip3Aa-S164P but not from Vip3Aa-S164A (Figure 2a). For the midgut proteases-processed Vip3Aa proteins, the band of protein complex 3 was observed from Vip3Aa-WT and Vip3Aa-S164T but not from Vip3Aa-S164A and Vip3Aa-S164P (Figure 2b). To estimate the molecular weight of the three protein complexes, the trypsin-processed Vip3Aa was analyzed by native SDS-PAGE. Two clear bands at ~240 kDa and ~200 kDa were observed (Figure 2c). The two bands in Figure 2c are assumed to the relatively dominant protein complex 1 and 2 in Figure 2a, and the molecular weight of complex 3 is predicted to be >240 kDa.

### 2.4. Composition of the Three Protein Complexes Formed from Vip3Aa Toxins after Tryptic Processing

To analyze the compositions of the three protein complexes, the bands corresponding to protein complexes 1, 2 and 3 (Figure 2a) were excised from the native PAGE gel (Figure 2a). The gel slices were mixed with SDS-PAGE sample buffer, heat denatured and loaded on SDS-PAGE gel to separate the proteins. All protein complexes contained the 65 kDa major fragment and multiple weak bands from 29 kDa to 66.4 kDa (Figure 3). Difference in composition was observed between the wild type and the mutants in the fragments below 20 kDa. A 19 kDa fragment was observed from protein complex 3 of trypsin-processed Vip3Aa-WT and Vip3Aa-S164T (Figure 3a). In comparison with the 19 kDa fragment, a slightly smaller fragment (17 kDa) was observed from the protein complex 3 from Vip3Aa-S164P (Figure 3a). An even smaller 15 kDa fragment was observed from protein complex 1 of Vip3Aa-WT and three mutants (Figure 3b). No protein bands below 20 kDa were observed from the protein complex 2 (Figure 3b). In addition, a peptide showing molecular weight around 95 kDa was observed from the protein complex 1 and 3 but not from the protein complex 2 (Figure 3).

### 2.5. Identification of Tryptic Fragments from the 15, 17 and 19 kDa Protein Fragments by Peptide Fingerprinting

The 15 and 19 kDa fragments, isolated from protein complexes 1 and 3 of Vip3Aa-WT and the 17 kDa fragment from protein complexes 3 of Vip3Aa-S164P were analyzed by nano LC-MS/MS to identify the protein fragments. The identified peptides derived from the 15, 17 and 19 kDa fragments were mapped to the amino acid sequence from D32 to K195 of Vip3Aa protein, located at the N terminal region (Figure 4).

### 2.6. Correlation of Toxicity of Vip3Aa Protein with the Formation of the Protein Complex 3 Composed of 19 kDa and 65 kDa Peptides

Trypsin-processed Vip3Aa-WT, Vip3Aa-S164T, Vip3Aa-S164A and Vip3Aa-S164P were fed to the neonates of *S. litura,* respectively, to assay their insecticidal activity. After 96 h, 100% mortality was observed by feeding *S. litura* larvae with 5 μg/mL and 50 μg/mL of trypsin-processed Vip3Aa-WT or Vip3Aa-S164T. In contrast, neither trypsin-processed Vip3Aa-S164A nor Vip3Aa-S164P showed significant toxicity to the larvae of *S. litura* (Figure 5). The insecticidal activity of trypsin treated Vip3Aa proteins was consistent to that of Vip3Aa protoxins (Table 1) and correlated with the formation of the protein complex 3 composed of the 19 kDa and the 65 kDa peptides (Figure 2a and Figure 3a)

## 3. Discussion

Previous studies have indicated that proteolytic processing of Vip3A proteins in insect midgut is a key step to exert the insecticidal activity [3,13]. In insect midgut, Vip3A proteins are processed by midgut proteases to produce a 62–66 kDa protease resistant toxic core from the C-terminal part of Vip3A. However, a recent study indicated that deletion of the first 198 residues at the N-terminus outside the ~65 kDa fragment region could lead to a complete loss of insecticidal activity and the resulting Vip3Aa fragments became sensitive to trypsin degradation [28]. Current studies have also shown that with treatment of Vip3A by trypsin, a 19~20 kDa peptide from the N-terminal region could bind with the C-terminal 62~65 kDa peptide, leading to the formation of a ~360 kDa homo-tetramer, which was tolerant to degradation in the protease-rich environment [29]. This 19~20 kDa peptide was proposed to play a functional role in protecting the 62~65 kDa peptide from proteolytic degradation and is necessary for the toxicity of the toxin in insects [30]. The K152 to E168, included in the N terminus 19~20 kDa peptide of Vip3Aa, were predicted to be a loop structure by the three-dimensional structure modeling software LOMETS (https://zhanglab.ccmb.med.umich.edu/LOMETS/). Significant decrease of toxicity of Vip3Af in *Spodoptera frugiperda* and *Agrotis segetum* was observed after substitution of T167 or E168 by Ala [14,20]. S164 was considered to be a polar amino acid located at the C terminus of the K152-E168 loop. Both K152 (carrying a basic polar side chain) and D154 (carrying an acidic polar side chain) were predicted to be at the N terminus of the K152-E168 loop. Consequently, in this study, we chose K152, D154 and S164 as our targets to analyze potential effects on the toxicity of Vip3Aa after substitution of these three amino acids respectively by Ala. Results of bioassay showed that only substitutions at S164 affect the toxicity of Vip3Aa in *S. litura*. Three main protein complexes were observed in protease treated Vip3Aa-WT and its three S164 mutants by native PAGE (Figure 2a,b). The protein complexes were composed of protein fragments of 19 and 65 kDa, 17 and 65 kDa, 15 and 65 kDa, or a single 65 kDa peptide only, respectively (Figure 3). The 95 kDa protein band was observed in protein complexes 1 and 3 but not protein complex 2 from each Vip3Aa protein. We interpret that the 95 kDa fragment is complexed with a 15 kDa peptide in protein complex 1 or a 19 kDa peptide in protein complex 3 with a 65 kDa peptide (Figure 3). LC-MS/MS analysis of the peptides between 15~19 KDa from the complexes indicated that the 15 kDa, 17 kDa and 19 kDa peptides were all from the N terminus of Vip3Aa protein, which corresponds to the previous reported domain I in Vip3Af protein [21]. Vip3Aa protoxin could be processed in vivo by the midgut proteases of *S. litura*. Toxicity of the Vip3Aa toxin, pretreated by midgut proteases, in *S. litura* should be similar to that of Vip3Aa protoxins. In order to build the relationship between the toxicity and the presence of the protein complex 3, mortality of *S. litura* fed with trypsin-processed Vip3Aa toxin was calculated and compared to the bioassay results by feeding *S. litura* with Vip3Aa protoxins (Table 1). Bioassay results showed that the trypsin-processed Vip3Aa-WT and Vip3Aa-S164T, in which the protein complex 3 was formed (composed of 19 and 65 kDa peptides), had significant toxicity in larvae of *S. litura* while the trypsin-processed Vip3Aa-S164A and Vip3Aa-S164P did not form the protein complex 3 with the 19 and 65 kDa peptides and completely lost the toxicity (Figure 5). These results are corresponding to the results of bioassay using Vip3Aa protoxins (Table 1). The toxicity of Vip3Af against *S. frugiperda* and *A. segetum* has been suggested to correlate with the transient formation of a tetramer composed of 20 kDa and 62 kDa peptides before the final processing to smaller fragments [14,21]. In this study, 100% mortality was observed in *S. litura* larvae fed with tryptic Vip3Aa-WT or Vip3Aa-S164T, while <10% mortality was observed in *S. litura* larvae fed with tryptic Vip3Aa-S164A or Vip3Aa-S164P (Figure 5). The protein complex 3 (composed of the 19 and 65 kDa fragments but not the 17 and 65 kDa fragments) could only be observed in tryptic Vip3Aa-WT or Vip3Aa-S164T (Figure 3). Consequently, formation of the protein complex 3 from Vip3Aa-WT and Vip3Aa-S164T was directly correlated to the toxicity of Vip3Aa in *S. litura*. A protein complex showing closed molecular weight to the protein complex 3 in Vip3Aa-WT and Vip3Aa-S164T was observed from tryptic Vip3Aa-S164P (Figure 2a) but not observed from midgut proteases-processed Vip3Aa-S164P (Figure 2b). Composition of this protein complex was identified to be the 17 kDa and the 65 kDa fragments (Figure 3a), different from the complex 3 from Vip3Aa-WT and Vip3Aa-S164T (Figure 4). This protein complex was degraded after treatment of Vip3Aa-S164P with midgut proteases of *S. litura*, while the protein complex 3 from Vip3Aa-WT and Vip3Aa-S164T remained stable (Figure 2b). Both the 17 kDa and the 19 kDa peptides were identified from the N terminal 199 amino acid residues of Vip3Aa proteins (Figure 4). These results indicated that the complete 19 kDa fragment is essential for the stability of the protein complex 3 which correlated to the insecticidal activity of Vip3Aa toxin.

The oligomer of Vip3A formed after proteolytic processing was observed through gel filtration chromatography analysis [14,21,29,31]. In this study, three protein complexes were observed from Vip3Aa-WT and its three mutants by native PAGE (Figure 2). Molecular weights of protein complexes 1 and 2 observed in the native SDS-PAGE gel were predicted to be ~240 kDa and ~200 kDa (Figure 2c). It is interesting that the protein complex 3 from tryptic Vip3Aa-WT could not be observed in the native SDS-PAGE. Previous studies identified a further degradation of Vip3A proteins due to the introduction of the secondary cleavage sites after treating with SDS contained SDS-PAGE sample buffer [17]. We speculate that protein complex 3 is easy to be disassembled in the native SDS-PAGE. Due to the disappearance of the less abundant protein complex 3 in the native SDS-PAGE gel, its molecular weight could only be predicted to be >240 kDa. From the toxin Vip3Af, a ~360 kDa homo-tetramer composed of 20 kDa and 62 kDa peptides has been proposed to correlate with the toxicity of Vip3Af toxin [14,21]. In this study, Vip3Aa-WT and Vip3Aa-S164T were observed to form the complex 3 composed of the 19 and 65 kDa peptides (Figure 3a) and have toxicity (Figure 5). It is possible that the protein complex 3 from Vip3Aa-WT and Vip3Aa-S164T corresponds to the previously reported 360 kDa homo-tetramer from Vip3Af, which was predicted to be formed by four 85–90 kDa protein complexes, each of them was composed of a 19 kDa peptide and a 65 kDa peptide [14,29]. SDS-PAGE analysis of trypsin- or midgut proteases-processed Vip3Aa proteins showed nearly the same protein fragment patterns below 20 kDa from Vip3Aa-WT and three S164 mutants (Figure 1). However, the protein complex 3 composed of the 19 kDa and 65 kDa fragments could only be observed from Vip3Aa-WT and Vip3Aa-S164T (Figure 2a,b). This is because the 15 kDa, 17 kDa and 19 kDa fragments were all identified from the N terminus of Vip3Aa (Figure 4). The S164 is critical for the formation of the protein complex 3, composed of 19 kDa and 65 kDa peptides from Vip3Aa proteins.

In conclusion, the present study identified a > 240 kDa protein complex composed of the 19 kDa and 65 kDa fragments from Vip3Aa after proteolytic processing. The formation of this protein complex was determined to correlate with the toxicity of Vip3Aa in *S. litura* larvae. The S164 in Vip3Aa is critical for the formation of the >240 kDa protein complex and consequently the insecticidal activity. The results from this study provided new information on the insecticidal mechanism of Vip3Aa toxins.

## 4. Materials and Methods 

### 4.1. Site Directed Mutation on the vip3Aa Gene

The *vip3Aa* gene (NCBI Accession No. AF500478.2) was cloned from plasmid of Bt WB7, a native strain isolated from soil collected in Wuyi mountain (Fujian, China) [32], by the use of primer pairs P-3Aa-F and P-3Aa-R (Table 2). The pGEX-KG vector [33] was used for the heterologous expression of the *vip3Aa* gene in *E. coli* BL21 (DE3). To substitute nucleotides coding for a single amino acid residue in Vip3Aa protein, primer pairs containing the site substitutions in *vip3Aa* gene was carried out by PCR using the pGEX-Vip3Aa as the template. Primers for the site directed mutation were listed in Table 2. To replace the S164 by an Ala in Vip3Aa, primers P-3Aa-F and P-164A-R were used as the primer pair for the 1st round PCR to obtain the N-terminal part of *vip3Aa* gene. P-164A-F and P-3Aa-R were used as the primer pair for the 2nd round PCR to obtain the C-terminal part of *vip3Aa* gene. Both PCR products were diluted 1000 folds in water and used as the template for the 3rd round PCR using P-3Aa-F and P-3Aa-R as the primer pair to obtain the full-length *vip3Aa* gene with the codons coding for S164 replaced by codons coding for A164. PCR reactions were performed using the iProof^TM^ High-Fidelity Master Mix DNA polymerase (Bio-RAD, Hercules, CA, USA). Other site substituted mutants of *vip3Aa* gene were generated according to the same procedure by the use of corresponding primers (Table 2). The final products from the 3rd round PCR were purified and digested with restricted enzymes of *Nco* I and *Sac* I. Digested products were purified and ligated with pGEX-KG plasmid linearized with *Nco* I and *Sac* I. Plasmids carrying mutated *vip3Aa* gene were transformed into *E. coli* BL21 (DE3) cells for protein preparations.

### 4.2. Expression and Purification of Vip3Aa Proteins

To prepare Vip3Aa proteins, 250 µL of overnight culture of BL21 cells carrying a plasmid of pGEX-Vip3Aa were inoculated to 250 mL of LB in a 1 L flask. The bacterial cultures were incubated at 37 °C and shaken at 150 rpm until OD_600_ reached 0.5. Protein expression was induced by addition of 0.8 mM IPTG (Isopropyl β-D-thiogalactoside) to the cultures, followed by incubation at 16 °C for 24 h. The *E. coli* cells were then harvested by centrifugation at 14,000× *g* for 1 min and the cell pellets were resuspended and washed in GST binding buffer (PBS, pH 7.3). The cell suspension was sonicated with a sonicator (VC-50, Sonics & Materials Inc. Danbury, CT, USA), followed by centrifugation at 21,000× *g* for 10 min to pellet the cell debris. The supernatant containing soluble GST-Vip3Aa proteins was loaded onto a Glutathione Sepharose column. Purification of GST fusion proteins and removing of GST tag by thrombin followed the standard purification procedure described by manufacturer of Glutathione Sepharose 4B (GE healthcare, Chicago, IL, USA). All purification steps were conducted at 4 °C or on ice. The purified GST-Vip3Aa fusions or thrombin treated Vip3Aa were quantified by the Bradford method [34].

### 4.3. Insects Rearing and Bioassays

An inbred colony of *S. litura* reared in the laboratory for over 3 years (~30 generations) was used in this study. The *S. litura* colony was maintained on a soybean-based artificial diet at 27°C with 50% humidity and a photoperiod 16 h of light and 8 h of darkness.

Bioassays were conducted by diet overlay method [35]. GST tag of Vip3Aa-WT was removed while other Vip3Aa mutants were prepared as GST-Vip3Aa fusions for bioassays. Briefly, the Vip3Aa-WT toxins or GST-Vip3Aa fusions in a series of dilutions were prepared in water. A 200 μL aliquot of diluted toxin was overlaid on the surface (~7 cm^2^) of diet in each cup (30 mL plastic cup containing ~5 mL diet). Each concentration included 5 replications. Ten neonates were placed into each cup. Cups were covered with lids and kept in the rearing room at 27 °C, 50% humidity and a photoperiod of 16:8 (light:dark) for at least 96 h. Mortality of larvae in each cup was recorded every 24 h. Cups contain diet overlaid by 100 μg/mL or 250 μg/mL of GST tag protein diluted in water were prepared as negative controls. Probit analysis of the bioassay data was carried out using the POLO program [36] to estimate the LC_50_ and 95% confidence limits (CL). For bioassays using tryptic Vip3Aa proteins, Glutathione Sepharose carrying GST-Vip3Aa fusions were digested with 10 mg/mL trypsin. Tryptic Vip3Aa proteins were quantified by the Bradford method and diluted to the concentration of 5 μg/mL and 50 μg/mL for the diet overlay bioassays described above. Mortality of *S. litura* larvae fed with tryptic Vip3Aa proteins were recorded in 96 h and analyzed by Prism (version 8.2.0, GraphPad Software, San Diego, CA, USA).

### 4.4. In Vitro Proteolytic Processing of Vip3Aa Proteins

To prepare midgut proteases of *S. litura* larvae, mid-fifth-instar larvae of *S. litura* were immobilized on ice for several minutes and dissected to isolate the complete midgut without loss of its contents. Midgut homogenates were prepared by thorough homogenization of 5 midguts in a 1.5 mL microcentrifuge tube. Grounded tissues were centrifugated at 16,000× *g* for 10 min at 4 °C. The supernatant was collected and distributed into small aliquots, snap frozen in liquid nitrogen and then stored at −80 °C until use. The protein concentration of midgut protease preparations was measured using the Bradford method.

Affinity-purified Vip3Aa proteins were subjected to in vitro proteolytic processing with trypsin (Sigma-Aldrich Inc. St. Louis, MO, USA) or midgut proteases of *S. litura*. Vip3Aa fusions were incubated with 10 mg/mL of trypsin or 400 μg/mL midgut protease preparation in Tris-HCl buffer (20 mM Tris-HCl, 0.15 M NaCl, 5 mM EDTA, pH 8.6) at the ratio of 120:100 (trypsin:Vip3Aa, *w*/*w*) for the trypsin treatment and 40:100 (midgut protease:Vip3Aa, *w*/*w*) for the midgut protease treatment. In vitro digestion was carried out at 30 °C for 6 h.

### 4.5. Analysis of Vip3Aa Proteins by the Native Gel and SDS-PAGE Gel

Protease treated Vip3Aa proteins were immediately analyzed by native PAGE or SDS-PAGE. To analyze protein complexes of Vip3Aa by native PAGE, trypsin- or midgut proteases-processed Vip3Aa proteins were mixed with the 2 × native PAGE sample buffer (0.5 M Tris-HCl pH 6.8, 5% bromophenol blue, 30% glycerol) and separated by the electrophoretic analysis on a 6% native PAGE gel. To analyze denatured Vip3Aa proteins by SDS-PAGE, the processed Vip3Aa proteins or protein complexes excised from the native PAGE gel were mixed with 5 × SDS-PAGE sample buffer (0.2 M Tris-HCl pH 6.8, 1 M sucrose, 5 mM EDTA, 0.1% bromophenol blue, 10% SDS and 5% β-mercaptoethanol), heated at 99 °C for 10 min and centrifugated at 15,000× *g* for 5 min. The supernatant was loaded to a 10% SDS-PAGE gel for the electrophoretic analysis. To estimate the molecular weight of protein complexes of Vip3Aa proteins by native SDS-PAGE, trypsin-processed Vip3Aa proteins were mixed with 5 × SDS-PAGE gel sample buffer with the absence of β-mercaptoethanol and loaded to a 6% SDS-PAGE gel for the electrophoretic analysis.

### 4.6. Identification of Trypsin-Processed Fragments

Protein bands of the 19 kDa fragment and the 15 kDa fragment contained by the protein complexes of trypsin treated wild-type Vip3Aa and a protein band of the 17 kDa fragment contained by the protein complex from trypsin treated Vip3Aa-S164P were excised from the SDS-PAGE gel after staining with Coomassie blue and detained with 30% acetonitrile and 100 mM NH_4_HCO_3_. The gel slices were dried in a vacuum centrifuge, and the proteins were reduced in-gel with dithiothreitol (10 mM DTT and 100 mM NH_4_HCO_3_) for 30 min at 56 °C, then alkylated with iodoacetamide (200 mM IAA and 100 mM NH_4_HCO_3_) in dark at room temperature for 30 min. Gel slices were briefly rinsed with 100 mM NH_4_HCO_3_ and acetonitrile, respectively, followed by digestion with 12.5 ng/μL trypsin in 25 mM NH_4_HCO_3_ overnight. The peptides were extracted three times with 60% acetonitrile and 0.1% trifluoroacetic acid. The extracts were pooled and dried completely in a vacuum centrifuge. The peptide mass and sequence were determined by Liquid Chromatography (LC)—Electrospray Ionization (ESI) Tandem mass spectrometry (MS/MS) in a Q Exactive mass spectrometer which was coupled to Easy nLC (Proxeon Biosystems, Thermo Fisher Scientific, Shanghai, China). The MS data were analyzed using Max Quant (version 1.6.4.0, Max Planck Institute of Biochemistry, Munich, Germany) by searching the data against the amino acid sequence of Vip3Aa, and the intensity of sequenced peptide in the target protein was calculated.

## Figures and Tables

**Figure 1 toxins-12-00274-f001:**
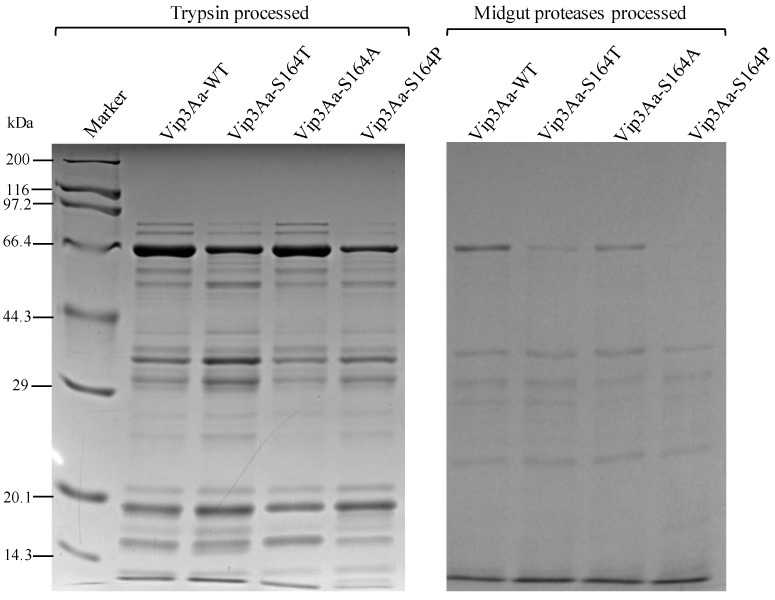
Analysis of Vip3Aa proteins after treatment by trypsin or midgut proteases of *S. litura.* Purified Vip3Aa-WT, Vip3Aa-S164T, Vip3Aa-S164A and Vip3Aa-S164P were *in vitro* digested by commercial trypsin or midgut proteases of *S. litura*. Processed proteins were mixed with 5 × SDS-PAGE sample buffer followed by heat denaturation and analyzed by the electrophoretic analysis in an SDS-PAGE gel.

**Figure 2 toxins-12-00274-f002:**
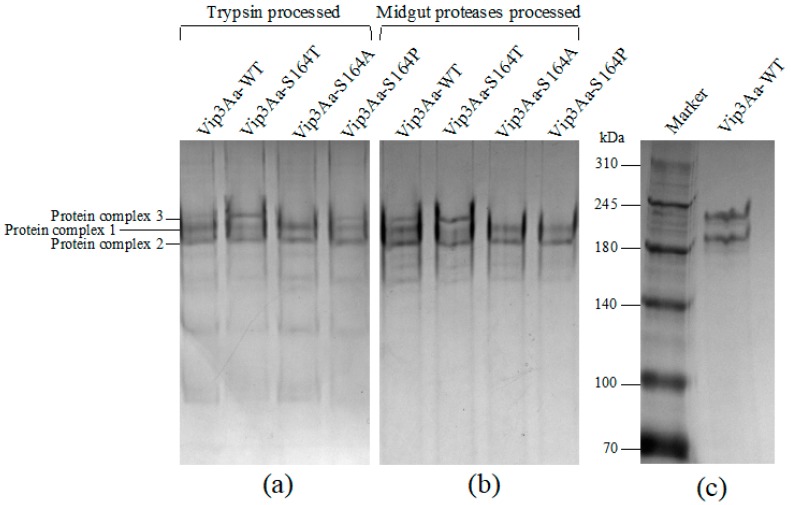
Analysis of native Vip3Aa proteins after proteolytic processing. Protease treated Vip3Aa-WT, Vip3Aa-S164T, Vip3Aa-S164A and Vip3Aa-S164P by either commercial trypsin or midgut proteases of *S. litura* larvae were analyzed by the electrophoretic analysis without heat denaturation. (**a**) Vip3Aa proteins after tryptic processing were analyzed in a native gel; (**b**): Vip3Aa proteins after processing by midgut proteases were analyzed in a native gel; (**c**): Vip3Aa proteins after tryptic processing were mixed with 5 × SDS-PAGE sample buffer without β-mercaptoethanol and analyzed in an SDS-PAGE gel. Protein complex 1, protein complex 2 and protein complex 3 in panel (**a**) indicate gel bands sliced from each lane in the native gel.

**Figure 3 toxins-12-00274-f003:**
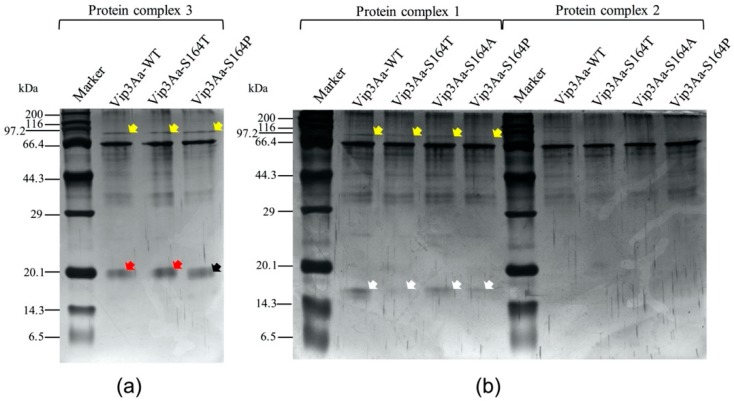
Separation of peptides from protein complexes of tryptic Vip3Aa proteins. Major protein bands representing different protein complexes in Figure 2a were sliced and separated in an SDS-PAGE gel. (**a**) peptides separated from the protein complex 3 in Figure 2a; (**b**) peptides separated from the protein complexes 1 and 2 respectively in Figure 2a. The yellow, red, black and white arrows indicate the bands of 95 kDa, 19 kDa, 17 kDa and 15 kDa fragments respectively.

**Figure 4 toxins-12-00274-f004:**
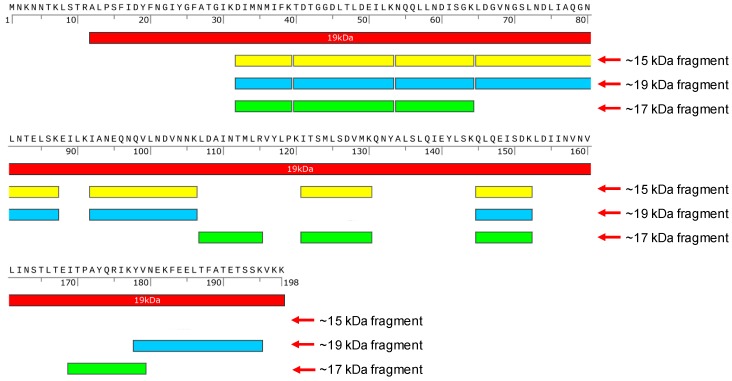
Schematic representation of the 15 kDa, 17 kDa and 19 kDa fragments from Vip3Aa protein. The first 198 amino acids at the N terminus of Vip3Aa were presented. The red box indicates amino acids corresponding to the N terminal 19 kDa fragment of Vip3Af. The yellow, blue and green boxes represent LC-MS/MS identified peptides from the 15 kDa, 19 kDa and 17 kDa fragments respectively.

**Figure 5 toxins-12-00274-f005:**
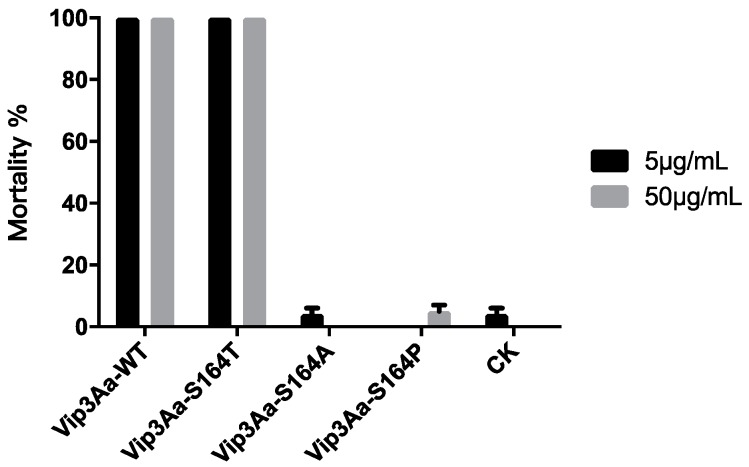
Mortality of *S. litura* larvae fed with trypsin-processed Vip3Aa proteins. After tryptic processing, Vip3Aa-WT, Vip3Aa-S164T, Vip3Aa-S164A and Vip3Aa-S164P were respectively fed to neonates of *S. litura* for 96 h to test their insecticidal activity. Error bars indicate the standard error of mortality among five replications.

**Table 1 toxins-12-00274-t001:** Insecticidal activity of Vip3Aa toxins in neonates of *S. litura.*

	LC_50_ (95% CI, μg/mL)	Slope	Χ^2^ (df, *p*)	Toxicity-Ratio
Vip3Aa-WT	1.69 (1.36–2.04)	3.50 ± 0.47	0.34 (4, 0.08)	1.0
GST-Vip3Aa-S164T	2.62 (2.24–3.17)	4.19 ± 0.47	8.88 (8, 1.11)	0.65
GST-Vip3Aa-S164A	>480	-	-	<0.0035
GST-Vip3Aa-S164P	>480	-	-	<0.0035
GST-Vip3Aa-K152A	2.23 (1.67–2.83)	3.16 ± 0.63	0.005 (3, 0.002)	0.76
GST-Vip3Aa-D154A	1.45 (1.36–1.55)	8.80 ± 0.99	2.74 (4, 0.69)	1.17

CI: confident interval.

**Table 2 toxins-12-00274-t002:** Primers for the site substitutions in *vip3Aa* gene.

Primers	Sequence 5′–3′
P-3Aa-F	CATGCCATGGACATGAACAAGAATAATACTAAAT
P-3Aa-R	CGAGCTCTTACTTAATAGAGACATCGT
P-164P-R	TTCAGTAAGTGTaggGTTAATAAGTACA
P-164P-F	ATGTACTTATTAACcctACACTTACTG
P-164T-R	TTCAGTAAGTGTggtGTTAATAAGTACA
P-164T-F	GTACTTATTAACaccACACTTACTGAAA
P-164A-F	ACTTATTAACgcgACACTTACTG
P-164A-R	AGTAAGTGTcgcGTTAATAAGTA
P-152A-F	GATTTCTGATgcgTTGGATATTA
P-152A-R	ATAATATCCAAcgcATCAGAAAT
P-154A-F	TGATAAGTTGgcgATTATTAATG
P-154A-R	ATTAATAATcgcCAACTTATCAG

Underlined sequences indicate the restricted enzyme sites of *Nco* I and *Sac* I. Lower case sequences indicate the mutant nucleotides in each primer.

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
