# Peer review of "Oligomer Formation and Insecticidal Activity of Bacillus thuringiensis Vip3Aa Toxin"

_toxins, 2020, doi:10.3390/toxins12040274_

Round 1
Reviewer 1 Report
The authors are trying to characterize Vip3A in regards to necessary oligomerization and a single point mutation that eliminates toxicity against Spodoptera litura. These results can provide additional information regarding Vip3A MOA. This reviewer applauds the authors for comparing both trypsin and midgut extract for proteolysis, but believes more studies are needed (described below). Although English is acceptable, there were a number of words that need correcting (not caught by Spellcheck) and several sentences that really did not make much sense.
Can the authors provide rationale as to why they only went after these three amino acid substitutions?
Comment: On lines 266-267: Are these really GST-Vip3A fusions if the GST is removed with thrombin? If not, please rephrase.
The major concern this reviewer has is in the lack of representation of midgut extract in this overall study. As there are a number of reports documenting differences in toxicity between trypsininzed and midgut extract digested Bt proteins, this reviewer would like to see the following studies conducted:
- A fig. 2 that was similar to Fig. 1 so that a direct comparison could be made between trypsin and midgut extract. In Fig.2, it looks like the gel was "cutoff" around 70 kDa or so that one could not observe the expected 65-66kDa bands that would have been at least one good comparison between the lower mol. wt. proteins for both trypsin and midgut extract. Also, this reviewer was trying compare the intensity of bands between Figs. 1 and 2. But without knowing the amount of protein loaded onto the gels, difficult to compare. Could the authors provide the amount of protein loaded onto all of the gels?
- Fig. 5 describes bioassays using trypsinized Vip3A. Could the authors provide similar data for Vip3A proteolyzed with midgut extract? or provide rationale as to why they did not conduct this?
Reviewer 2 Report
Dear Authors,
after revised the ms with title 'Oligomer formation and insecticidal activity of 2 Bacillus thuringiensis Vip3Aa toxin'. Overall it is a good works with interesting results.
So far so good but the in the M&M section is missing the stastical section and at results the in the mortality figure the statistical representation. Also in the Insects rearing and bioassays why do you use only neonate? The mortality every 24H and at the figure 5 you present what?
At discussion section lines 185-193 poorly discuss the mortality rate.
In the file i have some minor comments
My decision is major revision because th mortality section is not good and in the title of the ms you said about ΄insecticidal activity'

Reviewer 3 Report
In this manuscript the study of Vip toxin of Bacillus thuringiensis is described. Vip toxin family is an important insecticidal protein group but the mode of action of these toxins is not so well studied as for Cry toxins. The author have found that the 164 position in Vip3Aa protein is important for complex formation and is crucial for its insecticidal activity. Though the manuscript is interesting and provide important insights into the mode of action of Vip toxins, still there are some points, which should be clarified.
Major comments
- It is poorly described why the authors have chosen the 164 position. If it is a part of the big screening, it is better to include the results of that screening in the manuscript.
- It is not clear what is the purpose of Fig 2c section. What is the difference with Fig. 2a section and why is there no protein complex 3?
- According to the Fig 3, each protein complex presented at Fig 2, consists of dozen of smaller fragments. It seems to be very strange that such fragmented complexes remained stable during the native SDS-PAGE. Did the authors check that no protein degradation occured during the procedures described in section 2.4?
- Is not clear why protein complexes treated with trypsin but not native proteases, have been chosen for detailed analysis?
- Also, the procedure of trypsin digestion for 2.3 section, is not described in Methods.
- Though MS/MS analysis has been performed for the short fragments of complexes, it does not explain the structural difference between them. and this part of the work is poorly discussed in the Discussion. Also, it is not clear how the complexes are formed. I would suggest making native gel-electrophoresis for untreated complexes to determine the number of protein molecules in the complexes.
- In section 2.5 it is written that 15 and 19 kDa fragments have been digested with trypsin. What is the purpose of this digestion, considering that these fragments are products of trypsin digestion?
- The work lacks the structural models of the complexes.
Round 2
Reviewer 1 Report
This is the second resubmission of a manuscript to evaluate S164 in the toxicity of Vip3A against Spodoptera litura. The authors have adequately addressed this reviewer's comments. However, for two of the reviewer's comments:
The first response regarding rationale for only evaluating these three particular amino acids
AND
The last response regarding the rationale for not conducting bioassays with midgut extracts:
This reviewer would like to see these statements in the Discussion.
Author Response
This is the second resubmission of a manuscript to evaluate S164 in the toxicity of Vip3A against Spodoptera litura. The authors have adequately addressed this reviewer's comments. However, for two of the reviewer's comments:
The first response regarding rationale for only evaluating these three particular amino acids
AND
The last response regarding the rationale for not conducting bioassays with midgut extracts:
This reviewer would like to see these statements in the Discussion.
Reply: Thank you for the suggestions. We added description about the reason of choosing these three amino acids in the discussion part. Please see line 183 to 194. We also added some sentence to explain why we did not do bioassays using midgut proteases pretreated Vip3Aa toxins. Please see line 205 to 210.
Reviewer 2 Report
Accept in present form
Author Response
Accept in present form
Reply: Thank you.
Reviewer 3 Report
I would like to thank authors of the manuscript for the detailed answers. The manuscript is fine, thourgh I would suggest adding the information about the positions from the answer for first comment to the main text of the manuscript.
As to the last comment, I understand, that solving structure of the protein is out of the scope of this manuscript, but some speculations would add to the Discussion section, considering that authors have the predicted sturcture of the protein.
Author Response
I would like to thank authors of the manuscript for the detailed answers. The manuscript is fine, thourgh I would suggest adding the information about the positions from the answer for first comment to the main text of the manuscript.
Reply: Thank you for your suggestions. We added description about the reason of choosing these three amino acids in the discussion part. Please see line 183 to 194.
As to the last comment, I understand, that solving structure of the protein is out of the scope of this manuscript, but some speculations would add to the Discussion section, considering that authors have the predicted sturcture of the protein.
Reply: Thank you for your suggestions. According to current resources, I think it is not easy to have an accurate prediction in the strucutre of protein complexes of Vip3Aa toxin. We have speculated that the protein complex 3 (composed by 19 and 65 kDa peptides) are correspondent to previously reported 360 kDa homo-tetramer (please see line 238 to 239), whose structure was predicted in previous studies (Palma et al., 2017). We added some description in the structure speculations in the discussion section. Please see line 244 to 245. In further studies, we will try to analyze the exact molecular weight of protein complex 3 to verify our predictions.